# Label conditioned segmentation

**Tianyu Ma**                                                                    TM478@CORNELL.EDU
**Benjamin C. Lee**                                                    BCL2004@MED.CORNELL.EDU
**Mert R. Sabuncu**                                                      MSABUNCU@CORNELL.EDU
*Cornell University, Cornell Tech, Weill Cornell Medicine*

## Abstract

Semantic segmentation is an important task in computer vision that is often tackled with convolutional neural networks (CNNs). A CNN learns to produce pixel-level predictions through training on pairs of images and their corresponding ground-truth segmentation labels. For segmentation tasks with multiple classes, the standard approach is to use a network that computes a multi-channel probabilistic segmentation map, with each channel representing one class. In applications where the image grid size (e.g., when it is a 3D volume) and/or the number of labels is relatively large, the standard (baseline) approach can become prohibitively expensive for our computational resources. In this paper, we propose a simple yet effective method to address this challenge. In our approach, the segmentation network produces a single-channel output, while being conditioned on a single class label, which determines the output class of the network. Our method, called label conditioned segmentation (LCS), can be used to segment images with a very large number of classes, which might be infeasible for the baseline approach. We also demonstrate in the experiments that label conditioning can improve the accuracy of a given backbone architecture. Finally, as we show in our results, an LCS model can produce previously unseen fine-grained labels during inference time, when only coarse labels were available during training. We provide our code here: https://github.com/tym002/Label-conditioned-segmentation

**Keywords:** Semantic Segmentation

## 1. Introduction

Segmenting structures in images is a crucial task with many valuable real-life applications. For example, in biomedical image analysis, an accurate segmentation of the regions of interest (ROI) is usually the first step for computer-aided diagnosis (Al-Antari et al., 2018, 2020). With the success of deep neural networks in computer vision, architectures such as U-Net(Ronneberger et al., 2015) and the fully convolutional network (FCN) (Long et al., 2015) have become standard models for many image segmentation tasks. In these networks, the input to the model is usually a single 2D or 3D image, and the output is a multi-channel probabilistic map on the image grid. Each of the output channels typically correspond to one label class (Goyal et al., 2017). Therefore, the memory requirement for segmentation networks increases linearly with the number of classes. For many biomedical image segmentation tasks, the input images are 3D volumes that are large in each dimension (Evan et al., 2020). Moreover, the total number of classes can also be large. For example, the number of ROIs in a whole-brain segmentation task can be as large as 100. Thus, for most modern GPUs, the memory constraint can be a major bottleneck for the network architecture design, and researchers often have to use a batch size of one and a small number of channels (Myronenko, 2018).

The heavy memory overhead can be the result of a large number of classes and/or image grid size. Some common methods that can handle large image sizes include reducing the image resolution (Poudel et al., 2019) and using image patches during training (Ma et al., 2021). However, these methods typically hurt performance. To handle a large number of classes, one can train separate models on subsets of classes. Such an approach comes at the cost of training and saving multiple models. Recently, memory-efficient training has gained attention, especially for semantic segmentation with a large number of classes. For example, (Jain et al., 2021) uses class embedding to greatly reduce the number of classes the model predicts.

In this paper, we propose label conditioned segmentation (LCS), which is a simple scheme that can be used with different backbone segmentation networks. An LCS model outputs a single-channel segmentation map regardless of how many classes are used for training. The output class is conditioned on an additional input provided to the model. In our implementation, the conditioned label is presented as a two-channels atlas by concatenating a binary segmentation map and the corresponding image. During training, we (randomly) cycle through all the different classes of labels. During inference, the different labels are sequentially fed into the network to produce segmentation maps for each class, which are then combined into a single multi-label segmentation.

One advantage of LCS is the memory efficiency due to the single-channel output. Because the size of the model output is independent of the number of target classes, our method can handle segmentation tasks with a large (e.g., 100) number of classes in a single model. For many biomedical image segmentation tasks, especially with large 3D input images, the memory constraint makes it infeasible to implement large architectures. LCS offers a clear advantage in this setting. However, as we demonstrate in our experiments, LCS can boost segmentation performance, even when the memory constraint is not a major issue, e.g. when the number of labels is relatively modest. This, we believe, is likely due to the parameter efficiency and increased expressivity from class-dependent activations in the network. That said, the performance gain afforded by LCS becomes more pronounced with increasing number of label classes.

The proposed LCS framework has an additional use case at inference time. Because the output class is determined by the class label of the atlas the model is conditioned on, one can, in theory, present a new segmentation label that was not present in the training dataset. As our experimental results show, an LCS model can produce a useful segmentation under this scenario. An example we focus on is when training labels are coarser than inference-time labels. While LCS can be viewed as a one-shot learning framework (Feyjie et al., 2020; Zhang et al., 2019), there are some crucial details that make our setting unique. First, during training we assume we have a rich set of label classes that are all present in each of the training images. At test time, the query labels can include all or a subset of these labels, which is our primary use case. As a secondary use case, we consider novel test-time labels that can be generated on the same atlas used during training. These test-time labels will often be related to the original training label set. For instance, we might have a structure that we treat as a single ROI during training, but at inference time we might be interested in dividing it into two sub-parts, each associated with its own semantic label. To our knowledge, LCS is the first segmentation method that offers this kind of flexibility.

## 2. Methods

### 2.1. Proposed Approach

Label conditioned segmentation (see Figure 1) is a general scheme that can be employed with any machine-learning-based segmentation model. Since convolutional architectures are widely used these days, this is what we focus on in this work. We employ a standard 3D U-Net (Ronneberger et al., 2015) as our baseline architecture, since it is currently the most widely used model for biomedical image segmentation. For LCS, the baseline architecture has two distinct properties. First, it accepts an input atlas that determines the label class the model is conditioned on. This atlas is a binary label map and a corresponding image, both sub-sampled to the appropriate grid size. The two-channels atlas is concatenated to the bottleneck of the U-Net. The second distinct property of the LCS is that the output is a single-channel probabilistic segmentation map whose class is determined by the atlas.

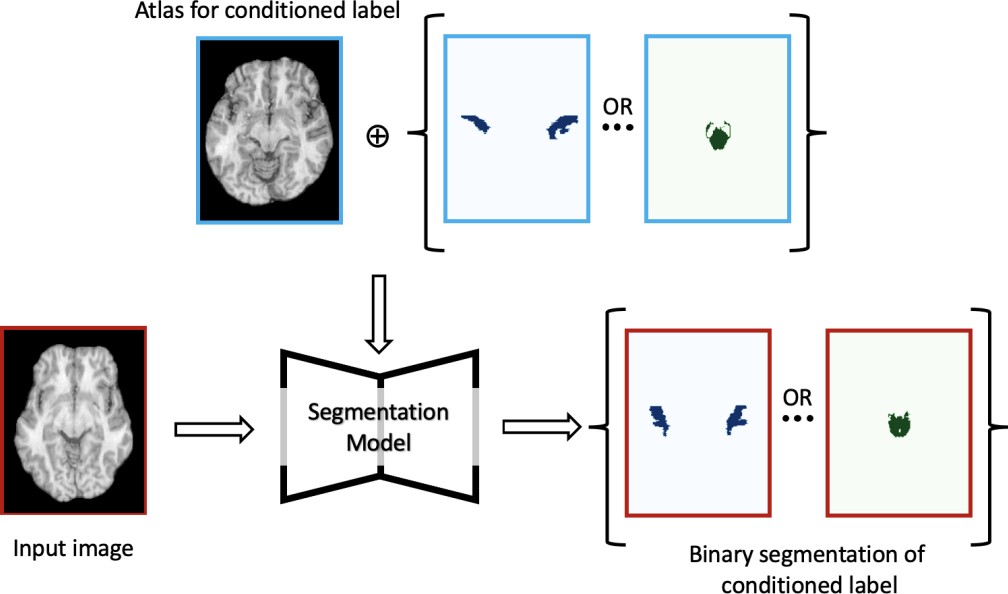

Figure 1: Label conditioned segmentation

### 2.2. Implementation Details

All convolution kernels in our backbone U-Net are $3\times3\times3$. There are 4 spatial scales, with a single convolution layer per scale. The highest spatial resolution has 8 output channels, and the number of channels doubles with each $2\times2\times2$ max-pool operation, in the encoder that ends with the bottleneck. In the decoder, each convolution layer is followed by a $2\times2\times2$ up-convolution.

For label conditioning, we randomly pick a training image-segmentation pair as the atlas. This atlas is used for both training and inference, thus can be considered as part of the model. In every training mini-batch, a random class label was chosen as the target

segmentation class. Furthermore, as a label augmentation strategy, with 50% probability we randomly chose a second class label, which was combined with the first label to create a merged target segmentation class. The atlas image and binary segmentation mask were then concatenated and down-sampled by a factor of 16 in each direction to feed into the bottleneck of the U-Net.

For the baseline, we used the same U-Net architecture, without the atlas input and the number of channels at the output equal to the number of segmentation labels.

The mini-batch size was set to 2 in all experiments. All models were optimized using soft-dice loss and the Adam optimizer (Kingma and Ba, 2014), with a learning rate of 0.0001 for 2000 epochs. In LCS, the loss function is computed only with respect to the label that the model was conditioned on.

We note that LCS and the baseline segmentation model share weights and activations across labels quite differently. For the baseline multi-channel output segmentation model, all weights and activations are common up to the penultimate layer. In our LCS model, however, while all network weights are shared across classes, the decoder can produce different activations for different label classes. We hypothesize that this property of LCS, in part, explains the improved accuracy we observe in our experiments.

### 2.3. Inference

For the baseline segmentation model, the output is a probabilistic segmentation map whose classes are determined by the training labels. Therefore, at inference time, the number of output channels and their corresponding semantic labels are pre-determined for the baseline model. For LCS, however, we are able to condition on any labels provided by the user at inference time. This label can be a training label or a previously unseen, novel label delineated on the atlas. In the default setting, we envision that LCS will be used to produce a segmentation for the training labels. The way this is done is, for each training label one forward pass is completed and the corresponding probabilistic map is saved. This process can be parallelized over multiple GPUs. Finally, all these probabilistic maps can be concatenated and normalized (to sum up to 1 at each voxel), which can be treated like a regular probabilistic segmentation.

Another possible use case for LCS is to leverage its generalization beyond training labels. For example, in many applications, it might be easier to obtain relatively large, coarse labels in the training examples. However, at inference time, we might be interested in obtaining fine-grained labels that correspond to sub-parts of the coarse labels. As we demonstrate in our experiments, LCS allows us to achieve this without the need for further training.

## 3. Experiments

We showcase our method on the Hammersmith dataset (Hammers et al., 2003; Wild et al., 2017) (http://brain-development.org/), which consists of 30 T1-weighted 3D brain MR scans from 30 healthy adults (ages between 20 to 54) with 95 manually delineated regions. We perform skull stripping and re-sample all scans to 1 mm isotropic voxel dimensions of $160 \times 208 \times 160$. We randomly split the 30 cases into 15 cases for training, 1 atlas (used to condition the LCS model), 5 cases for validation, and 9 cases for testing. We repeated the random split three times, with random atlas and non-overlapping test cases between splits.

### 3.1. Model Performance with Varying Number of Classes

We first trained a baseline and a label conditioned 3D U-Net on various selected brain structures. Because 10 classes is the largest number that our U-Net baseline can fit into one GPU, we hand-picked the following classes to focus on: hippocampus, amygdala, cerebellum, brainstem, caudate nucleus, putamen, thalamus, corpus callosum, lateral ventricle, and third ventricle. For both the baseline and LCS, we trained the models with 2, 6, 8, and 10 classes (first two, first four, etc.) of ground-truth labels.

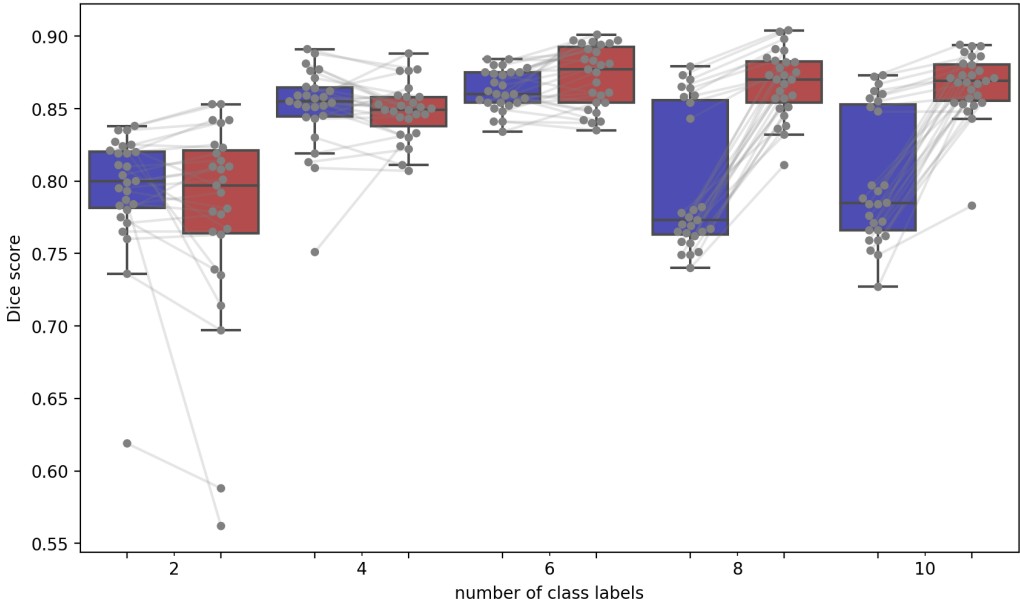

Figure 2: Distribution of test Dice scores for baseline (blue) and LCS (red) models, as the total number of output classes is varied. Each dot represents a test case. Same test cases are connected with a line for paired comparison.

The results are shown in Figure 2. When the number of classes is small (e.g. 2,4), the performance of LCS and the baseline is comparable (paired t-test p-value: 0.12 and 0.49). However, as the number of classes increases, we observe a significant Dice improvement for LCS (paired t-test p-values: 0.005 for 6 classes, $2e{-}10$ for 8 classes, and $7e{-}9$ for 10 classes). We hypothesize that the improvement in LCS is due to the fact that decoder activations can be different for different label classes, despite the shared parameters.

### 3.2. Segmenting a Large Number of Classes

With our current setup and implementation, the training-time RAM footprint of the baseline model with 1 channel output is about 5.8 GB per MRI scan in the mini-batch. The largest number of output channels our baseline U-Net can fit on a single GPU is 10. To fit all 95 output classes in our dataset, the baseline would require about 300% more memory.

Therefore, we need to train 10 different baseline U-Net models, each with 9-10 labels. For our LCS approach, on the other hand, we can train a single model for all labels.

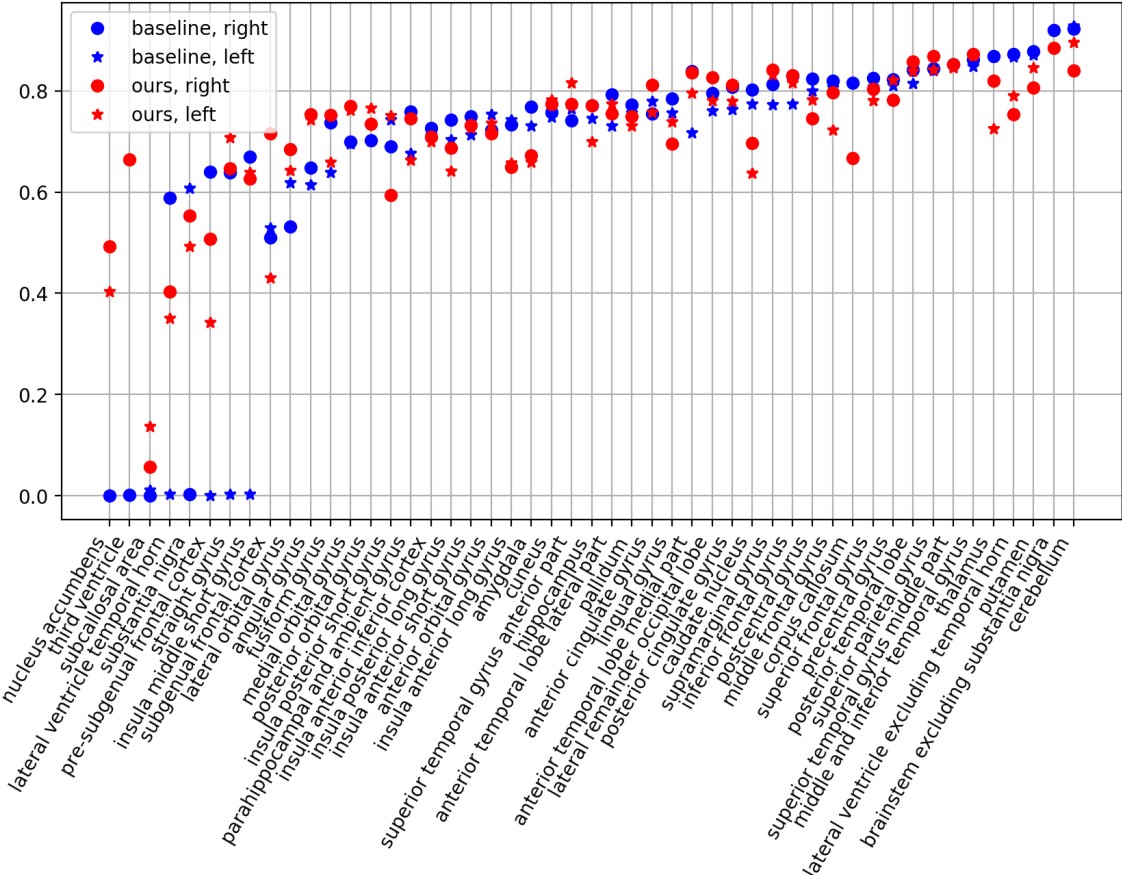

Figure 3: Dice overlaps of all 95 classes in the left and right hemispheres of the brain for baseline (blue) and LCS (red) models. Classes are in ascending order of the baseline performance.

Figure 3 shows the average Dice scores for all classes. Even though the baseline has 10 models and thus roughly ten times the capacity of LCS, our method achieves a better global average Dice score of 0.713, compared to 0.680 for the baseline. We notice that much of the performance boost of LCS is for the relatively smaller structures that the baseline U-Net struggles with. We emphasize that LCS has a memory overhead that is a tenth of the baseline.

### 3.3. Segmenting Novel Classes

In this section, we wanted to explore LCS's capability to produce segmentations for previously unseen labels. For our experiment, we merged some of the labels into larger structures

that represent a coarser segmentation. For example, the middle and anterior parts of the superior temporal gyrus were combined into a single ROI, the superior temporal gyrus, during training. During testing, we asked the model to predict the middle and anterior parts separately, by conditioning on the fine-grained label in the atlas. Figure 4, for instance, shows the visualization of the ground-truth temporal gyrus and the LCS model predictions in a test case. We observe that LCS can produce segmentation maps on the previously unseen sub-classes with acceptable accuracy.

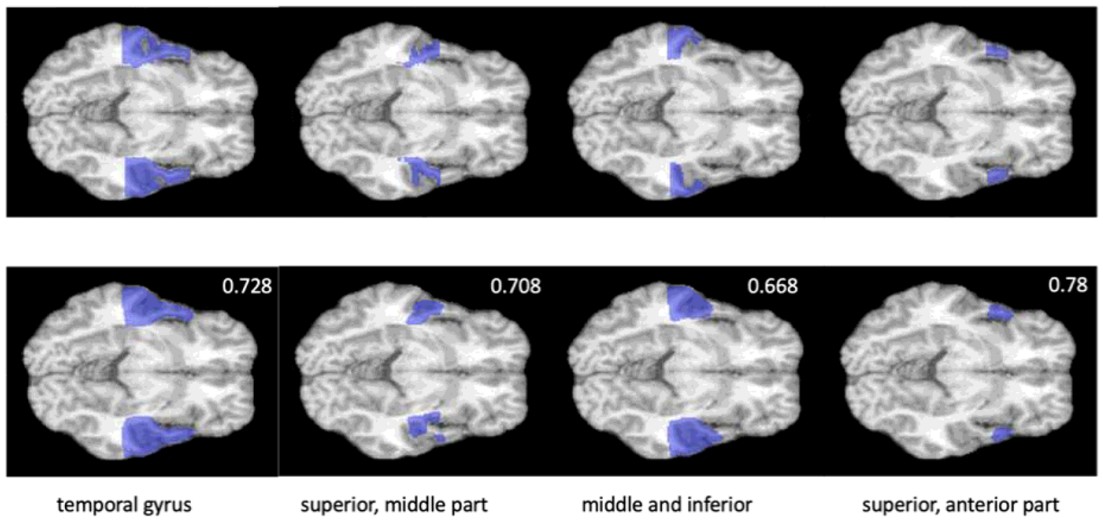

Figure 4: Visualization of ground-truth segmentations (top) and LCS predictions (bottom) of temporal gyrus along with their Dice overlaps. The LCS model is only trained with the whole temporal gyrus segmentation in the first column.

Table 1 provides quantitative results and lists the coarse training labels and fine-grained inference labels used in this experiment. As a point of comparison, we show (naive) Dice scores for the LCS predictions conditioned on the coarse training labels. We observe that the Dice scores for the LCS predictions conditioned on the previously unseen fine-grained labels are consistently higher than the naive baseline.

## 4. Conclusion

For many biomedical image segmentation tasks with large 3D data, it is often not feasible to train a single model for all classes. To address this limitation, we propose a novel scheme called label conditioned segmentation, or LCS. In LCS, in addition to the input image, the model accepts a label, which conditions the output. The single-channel output represents the binary segmentation corresponding to the conditioned label. In our implementation, we represent the conditioned label as an atlas image-segmentation pair, presented at the U-Net bottleneck.

Table 1: LCS results when training on coarse labels and testing on fine-grained labels. "Naive Dice" scores quantify the overlap between the LCS prediction conditioned on the coarse training label and the ground-truth fine-grained label. "Fine-grained Dice" is for the LCS prediction conditioned on the correct fine-grained atlas label, which was not seen during training.

| Coarse Training Label | Dice | Fine-Grained Label | Naive Dice | Fine-Grained Dice |
|---|---|---|---|---|
| anterior temporal lobe | 0.756 | medial part | 0.571 | 0.670 |
| | | lateral part | 0.452 | 0.590 |
| temporal gyrus | 0.886 | superior, middle part | 0.488 | 0.669 |
| | | superior, anterior part | 0.573 | 0.721 |
| | | middle and inferior | 0.196 | 0.644 |
| frontal gyrus | 0.888 | middle | 0.531 | 0.611 |
| | | inferior | 0.212 | 0.487 |
| | | superior | 0.514 | 0.728 |
| cingulate gyrus | 0.796 | anterior | 0.540 | 0.720 |
| | | posterior | 0.531 | 0.792 |
| orbital gyrus | 0.797 | medial | 0.389 | 0.589 |
| | | lateral | 0.179 | 0.595 |
| | | posterior | 0.335 | 0.388 |
| | | anterior | 0.405 | 0.508 |
| insula short gyrus | 0.789 | anterior | 0.481 | 0.524 |
| | | middle | 0.334 | 0.348 |
| | | posterior | 0.361 | 0.458 |

Our experiments indicate that LCS can produce better results than the baseline, even when the number of labels are not prohibitively high. We suspect that this is because the LCS model can have different activations in the hidden layers for different labels, which is not the case for the baseline approach. We also demonstrate that the proposed approach can be used to produce segmentations of fine-grained labels at inference time, despite being trained on coarse labels. We believe that this paper establishes the feasibility and value of LCS, but there are many open questions that will need to be explored. For instance, there might be better representations for the conditioned label and better ways to present it as an input to the model. Also, we are interested in further studying the limits of the generalization of LCS to previously unseen labels.

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

## Appendix A. Results for 49 Classes Segmentation

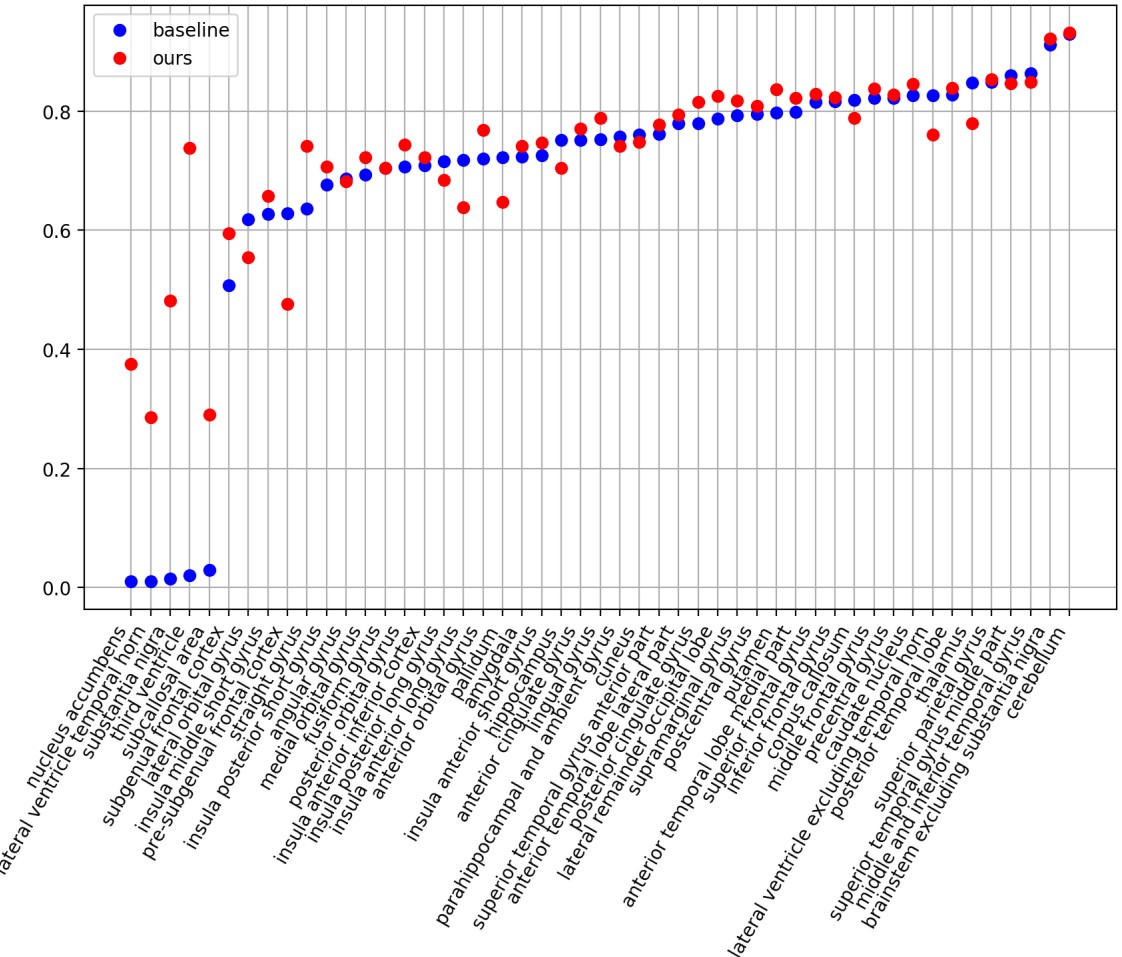

Figure 5: Dice overlaps of 49 classes for baseline (blue) and LCS (red) models. Classes are in ascending order of the baseline performance.

