# OpenReview forum: "Label conditioned segmentation"
_MIDL.io/2022/Conference — MIDL 2022_

### Official Review · Reviewer_Mz7J · 2022-01-22

**Confidence:** 4
**Preliminary Rating:** 4
**Recommendation:** Oral, Poster

**Summary:**

The authors propose label conditioned segmentation (LCS), which produces a single-channel output while being conditioned on a two-channel atlas (concatenation of binary label map and the corresponding example image). It can be used to segment images with a very large number of classes, and even improve the performance thanks to the parameter efficiency. Besides, such framework shows its superiority in segment more fine-grained classes which do not appear in the training.

**Strengths:**

-This paper tackles an important task of segmentation with a lot of classes in a memory-efficient way.

-The proposed method gain some insights from few-shot segmentation to introduce the image-label pair as the condition in the bottleneck for a label conditioned segmentation, which is novel in the task.

-The framework saves the computation resource on the one hand, on the other hand, it improves the performance when there are a lot of classes compared to the baseline model, which is impressive.

-The author further explores the possibility to conduct conditional segmentation with a more fine-grained label that does not appear in the training. This somehow indicates that the network is able to infer some spatial correlation from the condition pair (atlas).



**Weaknesses:**

While it is helpful to have Figure one for illustrating the basic idea. I found it not easy to grasp the idea at first glance, especially for the concept of atlas.  I would suggest that the author add a few sentences to briefly describe the concept.


-How does atlas integrate into the bottleneck? (A more detailed description in the Appendix may help)

-Are there any relationship between the input image and atlas? At first glance, the atlas looks very similar to a downsampled version of the input image, which makes it a little confusing that the network is just doing some refinement based on the coarse label for the input sample. It is until I read Section 3 that I start to understand that atlas is just the image-label pair from another sample rather than the input image (Can the authors confirm whether what I understand is correct?)


Considering that the network is able to conduct conditional segmentation with a more fine-grained label that does not appear in the training during inference, I would speculate that inputting the mask-pair atlas is important in achieving that. Nevertheless, I am wondering what may happen if the input is not the spatial atlas pair but other information (one-hot vector labels/ image and one-hot vector label pairs). Presumably, it won't be easy to have such generation ability to unknow class. Still, it would be more convincing to show how this influences the performance for segmentation compared to using atlas.


The authors mentioned that they "repeated the random split three times, with non-overlapping test cases between splits." I am wondering how the model is sensitive to the choice of the atlas sample. For example, keeping the test set fixed, just randomly choosing different cases to be the atlas, would it influence the model performance a lot? It appears to me that there is no detailed description of the consideration for atlas selection. Yet the atlas it self seems to be important for the model performance.




**Deanonymize Review:**

no

**Detailed Comments:**


Minor comments:

Description of Figure3 is a little confusing to me, as there appears right/left in the figure, but there is no description in the caption on the main text.

The description in section 2.3 is a little hard to understand. "For LCS, however, we are able to condition on any label at inference time. " From my understanding, the training label in this paragraph refers to the training class rather than the specific label used as the atlas, right?


Other comments:

The author mentioned that "better representations for the conditioned label and better ways to present it as an input to the model. " could be a future direction. I am thinking whether it would be an interesting direction to investigate how to encode the image-label atlas. For example, rather than encoding the image and label masl in the atlas directly in the bottleneck, mapping it to some hidden representation. Or using multiple samples to generate an atlas prototype rather than using just one example.





**Final Rating After The Rebuttal:**

5: Strong Accept

**Justification Of The Final Rating:**

Thanks for the reponse from the author. This paper is interesting and addressed an important topic in medical segmentation. It is also quite promising that they have the ability to conduct conditional segmentation with a more fine-grained label which does not appear in the training. I think it is also promising to explore the possible direction mentioned by the author. Therefore, I incline to accept this paper. Nevertheless, I do encourage the author to incorporate the suggestion from the Reviewer PqCB in the further version.


**Paper Type:**

both

**Questions To Address In The Rebuttal:**

See Weakness and Detailed Comments. Overall, this is an interesting paper, I recommend a weak acceptance as the initial rating. If the authors can address my concerns, I would consider raising the score.

**Special Issue:**

yes

---

### Official Review · Reviewer_PqCB · 2022-01-24

**Confidence:** 5
**Preliminary Rating:** 2
**Recommendation:** Poster

**Summary:**

The paper proposes a label conditioned segmentation method where the segmentation network produces a single-channel output and no need for multi-channel output which is expensive for computational resources multi-class segmentation tasks. The method is also able to predict previously unseen classes during inference time.

**Strengths:**

- The model doesn't need to predict multi-channel output for each class which may help for the dataset with a very large number of classes.

- The model can be conditioned for unseen classes which is quite interesting.


**Weaknesses:**

- How is it computationally efficient over a standard approach (each channel representing one class)? LCS required forming an atlas with two channels and additional processing parameters and computational resources in the model.

- In addition, it needs to run multiple times to get a segmentation for all the classes which is more expensive computationally in inference time

- Some classes may not exist in a test sample then how to decide atlas conditioning?

- LCS required a dedicated sample to form an atlas to use in both training and inference. How model behaves if a different atlas uses in the inference time?

- Atlas could be overfitted as it is used for both training and inference

- It would be great to compare computational speed to show the efficiency of the proposed method

- In general, do we have or need a medical segmentation dataset with a large number of classes?

- The dataset used to validate the proposed method is too small. “15 cases for training, 1 atlas (used to ​​condition the LCS model), 5 cases for validation, and 9 cases for testing”.


**Deanonymize Review:**

no

**Final Rating After The Rebuttal:**

3: Borderline

**Justification Of The Final Rating:**

Thanks for the response from the authors. The approach is helpful in terms of downstream segmentation tasks with a large number of segmentation classes. However, it seems the method is still needed to validate on different perspectives as below-
1. Computational speed and required memory w and w/o LCS  to predict all the classes.
2. Prediction performance with different atlas(es)

**Paper Type:**

both

**Questions To Address In The Rebuttal:**

- It would be great to compare computational speed to show the efficiency of the proposed method over the conventional segmentation method (multi-channel prediction for multi-class segmentation)

- LCS required dedicated scans to form atlas to use in both training and inference. How model behaves if a different atlas uses in the inference time?

**Special Issue:**

no

---

### Official Review · Reviewer_skJJ · 2022-01-25

**Confidence:** 5
**Preliminary Rating:** 5
**Recommendation:** Best Paper Award, Oral, Poster

**Summary:**

The authors propose a method to overcome the memory burden when segmenting large numbers of classes, especially in the case of 3D data. The proposed method takes as input to the bottleneck layer of a U-Net architecture an atlas which guides the network on which class to segment from the image. In this way, the network only produces a single output label (guided by the atlas) at a time rather than labels for all classes at once. Further, the authors show their approach can be used to "one-shot" segment similar class labels to those seen in training (more fine-grained).

**Strengths:**

1. The authors' proposed method address an important problem. The number of segmentation classes can quickly grow as datasets grow considerably larger and the scope of tasks for potential deep learning solutions grow as well. For 2D segmentation, this is not a significant issue as the number of parameters and size of intermediate representations scale fairly well (e.g. $512 \times 512 \times c$ output with $1 \times 1$ convolutional kernels applied to $64$ input feature maps would require $64 \times c$ parameters and $512 \times 512 \times c$ output values, given $c$ output classes). However as we move to 3D segmentation tasks such as medical scans or video data, those memory constraints become an issue fast (e.g. $160 \times 208 \times 160 \times c$ output with $1 \times 1 \times 1$ convolutional kernels applied to $64$ input feature maps would require $64 \times c$ parameters and $160 \times 208 \times 160 \times c$ output values, given $c$ output classes). So in our two scenarios (2D and 3D), given choices of $c = {1, 10, 50, 100}$ and assuming each value (parameters and feature map value) are stored as 32 bit numbers, we would have the following: 2D = {1, 10, 50, 100} MB and 3D = {20, 203, 1015, 2031} MB.

2. The authors discuss the performance improvements over traditional approaches. I believe this is coming two-fold. First, the atlas can be thought of as a shape prior, giving some context to the network about what sort of shape to expect and this will give some minor performance improvement. However, where the largest improvement I suspect is coming from is the multi-task nature of the problem and approach. Given a network to segment dogs and a network to segment cats, or one network which learns to segment dogs and cats, it's likely the second approach will perform better, because the cat labels actually provide some weak context information for the dog class and vice-versa. As the number of classes increase, this context becomes more useful. The authors even push this further by artificially introducing new classes that are "catdog"s where the segmentation masks for cats and dogs are combined into one big mask and then segmented by the network.

3. Places with large compute resources (e.g. gpu farms), can get a very nice parallelization benefit from this style of approach. Instead of one large model which outputs 100 classes. Users can send the reduced model to 100 GPUs, each predicting a single class, then combine the results. This would produce a fairly nice speed-up at places such as hospitals which have to process thousands of scans very quickly. The flexability to only test on some classes also helps speed this up even further.

4. The impact of this work stretches far beyond its proposed applications. Segmentation is a general problem in computer vision and methods for fast and low memory models (e.g. autonomous vehicles) which can segment a large number of objects. There are immediate extensions that come to mind with extending this approach to areas such as instance and panoptic segmentation tasks, perhaps in the open world setting where maybe the object being segmented doesn't match anything in the atlas and can be added as a novel class (the authors reference a similar one-shot learning case).

**Weaknesses:**

1. Related to strength point 2, having additional classes in a network provides context clues. Therefore when randomly dividing up the 95 classes into sets of 9-10, the performance reported for each class might change (possibly fairly significantly) depending on what other classes it happened to be grouped with. In an ideal scenario, it would have been nice to see here some random shuffling of these groupings and the experiments repeated say 4-5 times then have error bars showing that variation.

2. With the atlas being the only clue to the network as to which class it needs to segment, and the atlas containing a segmentation mask which has been downsampled by a factor of 16, there is a real concern that segmenting very small structures with such an approach would be impossible. See detailed comment about this.

**Deanonymize Review:**

yes

**Detailed Comments:**

1. Related to strengths point number 1. I would have really like to see a small table comparing the memory (vRAM) impact of 100 classes in the normal segmentation setting vs in the proposed setting.

2. Related to strengths point number 2. If the authors agree with the points made in this numbered comment, it would be nice to have some discussion of this in the paper for others to better understand where these performance gains are [possibly] coming from. The hypothesized explanation provided the decoder "can be different" is, in the reviewer's opinion, not a strong hypothesis about where the improvement is coming from.

3. Related to weakness 2. Have the authors considered this case? Have they tried any other possible arrangements such as having the atlas concatenated to the feature maps at every layer of the encoder/decoder? By giving the atlas to the network earlier, this would allow the encoder to extract class specific features (similar to query-guided proposals in person re-identification) rather than being forced to stay general in the encoder and only pick out class specifics in the decoder.

4. This statement "...that encoder activations can be different for different label classes, despite the shared parameters" is incorrect. The authors earlier statement of "while all network weights are shared across classes, the decoder can produce different activations for different label classes." is correct. The decoder can be different, the encoder cannot as the encoder doesn't see what the class label is until the bottleneck.

**Final Rating After The Rebuttal:**

5: Strong Accept

**Justification Of The Final Rating:**

My initial assessment of the paper has not changed. The authors are solving an important problem. There are significant potential extensions of this work to a number of other problems. I strongly am in support of acceptance.

**Paper Type:**

methodological development

**Questions To Address In The Rebuttal:**

No questions, just comments in the above sections that can be addressed to make the paper even better. But overall, the reviewer thinks this is a very strong work with implications and uses far beyond the applications shown.

**Special Issue:**

yes

---

### Meta-Review · Area_Chair_YNFG · 2022-02-16

**Recommendation:** Accept (Oral)
**Confidence:** 5

**Metareview:**

The paper received three detailed reviews, two of which are fairly enthusiastic. A recurrent concern that was not fully addressed pertains to the use and definition of a specific atlas which may introduce some limitations that could be investigated further. This, however, might be accomplished through a modification of discussion and does not affect overall rating. However, it might be of importance for future work.

---

### Decision · Program_Chairs · 2022-02-28

Accept